# Adaptive strategies of Egyptian crowfoot grass (*Dactyloctenium aegyptium* (L.) Willd): Germination responses to environmental factors

Ahmad Zare[ID]1*, Elham Elahifard1, Zahra Asadinejad2

1 Department of Plant Production and Genetics, Faculty of Agriculture, Agricultural Sciences and Natural Resources University of Khuzestan, Bavi, Mollasani, Iran, 2 M.Sc. Graduated in Weed Science, Department Plant Production and Genetics, Faculty of Agriculture, Agricultural Sciences and Natural Resources University of Khuzestan, Bavi, Mollasani, Khuzestan, Iran

* ahmadzare@asnrukh.ac.ir

## Abstract

Egyptian crowfoot grass (Dactyloctenium aegyptium (L.) Willd.) is a troublesome $C_4$ grass weed that competes with crops such as rice (Direct Seeded Rice), sugarcane and urban green space. The effects of constant and alternating temperatures, storage times, light, salinity, and water stress on germination of Egyptian crowfoot grass studied under laboratory conditions. Germination was stimulated by light, and seed germination percentage was higher under alternative temperatures compared to constant temperatures. Results also showed that germination percentage increased as the storage period lengthened. The highest germination percentage (97.33%) was observed under 30/20°C alternating temperature with light in seeds stored for 12 months. The time required to reach maximum germination was shorter under both constant and alternating temperatures for seeds stored for 12 months compared to those stored for 5 months and 2 months, respectively. Osmotic stress had moderate negative effects on germination, with 45% of the Egyptian crowfoot grass seeds able to germinate at −0.6 MPa,. Complete germination inhibition was observed at an osmotic potential of −1 MPa. The osmotic potential required for 50% inhibition of maximum germination was 0.49 MPa. Also results of salinity indicated that 23% of seeds germinated at sodium chloride (NaCl) concentration of 200 mM. A 50% reduction in germination of Egyptian crowfoot grass was caused 162 mM. To reduce the soil seed bank of this photoblastic weed, we recommend implementing deep tillage, applying straw mulch, and utilizing the false and stale seedbed techniques.

## 1. Introduction

Egyptian crowfoot grass (*Dactyloctenium aegyptium* (L.) Willd.) is a tufted, slightly stoloniferous annual or short-lived perennial grass that grows up to 75 cm in height [1]. It has been introduced as a C4 weed in 19 crops and 45 countries, [2] In different

**Data availability statement:** All relevant data are within the paper and its Supporting Information files.

**Funding:** This study was financially supported by the University of Khuzestan Agricultural Sciences and Natural Resources department in the form of a grant awarded to AZ (981/09). No additional external funding was received for this study. The funders had no role in study design, data collection and analysis, decision to publish, or preparation of the manuscript.

**Competing interests:** The authors have declared that no competing interests exist.

crops such as rice (*Oryza sativa* L*.*), cotton (*Gossypium hirsutum* L.), sugarcane (*Saccharum officinarum* L.), peanuts (*Arachis hypogaea* L.), corn (*Zea mays* L.), and vegetables, it is known to be a tenacious, and troublesome weed [3]. This species also grows in sunny or lightly shaded places in gardens, waste places, dikes, and embankments [4] The seed production ability is estimated at up to 66,000 seeds per plant, and propagates mainly due to the large production of seeds [3]. It is considered a weed in rice, sugarcane production regions and landscapes of southwestern Iran.

Germination and emergence of weed seeds are critical developmental stages that determine a weed's success in agro-ecosystems [5]. The timing of seedling emergence, determined by germination-triggering abiotic cues, shapes the initial environmental conditions they experience and critically influences plant survival and establishment [6]. These processes are regulated by environmental factors, including temperature, soil pH, salinity, water availability, and seed burial depth [7]. Previous research indicates that these factors can either initiate or inhibit seed germination. Notably, temperature is a primary determinant of germination when other factors are non-limiting, with its effects varying among species within the same genus [8]. Temperature critically regulates seed germination rates through its role in overcoming seed dormancy. Determining optimal germination temperatures facilitates the prediction of peak emergence events within a season, informing targeted weed management strategies [7].

Studies have demonstrated that light and alternating temperature regimes serve as two critical environmental factors stimulating seed germination [9,10]. The Photochrome absorption levels in a seed can be used to evaluate the competitive pressure in its immediate environment and early detection of competition before germination improves seedling survival rates [11]. In an environment with minimal competition, the seed senses a higher ratio of red light compared to infrared light, encouraging germination, Seeds sprouting in complete darkness may have elevated levels of far-red phytochromes in their embryonic tissue, which assist in sensing intense competition [12]. By determining that a weed is positively photoblastic, light-limiting management techniques, such as mulching with crop residues, can be applied [13].

Water stress can delay, reduce, or inhibit weed seed germination. However, the ability of certain weed species to germinate under low-moisture conditions provides a competitive advantage, enabling them to outperform many crops in germination and growth [7]. Salinity is a major environmental stressor that reduces germination rates, delays germination onset, and impairs seedling establishment, ultimately limiting agricultural productivity and sustainability in arid and semi-arid regions [14]. After-ripening refers to physiological and biochemical changes occurring in dry (non-imbibed) seeds following dispersal, which break seed dormancy and enhance germination capacity and After-ripening is significantly influenced by storage conditions and their duration [15]. The process may be attributed to Changes in seed chemical composition, decreased abscisic acid (ABA) levels during storage and increased Gibberellic acid concentrations [16]

Plentiful seed output is a hallmark of thriving weeds. A robust soil seedbank gives weeds an advantage over crops and native plants, particularly when they grow faster

than desired species. These substantial seed stores result in ongoing control challenges [17]. A thorough comprehension of the biological mechanisms behind seed germination, particularly under varying environmental conditions, is fundamental for developing weed management strategies. The ability of weed seeds to germinate and successfully emerge from the soil plays a pivotal role in their persistence and colonization within agroecosystems [18–20]. Understanding the relationship between environmental factors and plant physiology is crucial for analyzing and predicting ecological dynamics. This interaction is shaped by local climatic conditions, including latitude, elevation, soil moisture, temperature and precipitation patterns, light availability, and photoperiod (day length) [21]. Understanding seed dormancy, germination and its relationship with environmental factors is essential for predicting germination timing and implementing effective weed management strategies [22].

Weed seed populations exhibit significant divergence in germination behavior across heterogeneous geographic origins and habitat types [23,24]. Germination responses of Egyptian crowfoot grass populations exhibit significant geographic variation. While US populations germinate across 15–40°C peaking at 94% under 35/25 C alternating day/night temperatures [25], Philippine populations show ≤20% germination in darkness at 30/20C night [2]. African populations achieve 100% germination in both light and dark regimes [26]. In response to water stress, Egyptian crowfoot grass seeds could tolerate water stress up to −0.6 MPa, [2,25]. [27] reported that the Egyptian crowfoot grass population could survive with a low growth rate at salinities of up to 50% artificial seawater. Weed seeds may also have different germination requirements if they mature under different environmental conditions.

A key methodological distinction concerns post-harvest storage duration: 6 months for the US population versus 9 months for the Philippine population, aligned with respective maternal plant rainfall regimes prior to testing. Storage intervals (post-ripening) may significantly influence germination capacity and velocity, potentially altering optimal weed management timing [28].

The agricultural landscape of Khuzestan Province has undergone significant transformation due to climate change impacts, particularly increasing thermal regimes and diminishing hydrological resources. These environmental shifts have necessitated the adoption of direct-seeded rice methodologies as an alternative to traditional paddy-based cultivation. Consequently, there emerges a critical research imperative to elucidate the germination dynamics of autochthonous populations of Egyptian crowfoot grass, a competitive weed species exhibiting increasing proliferation in Iranian rice production systems.

This study expands on prior regional observations by quantifying germination plasticity in Iranian populations under projected climate scenarios. The objectives of this research were to determine (a) the effect of temperature and light in three storage times (2, 5 and 12 months after rainfall of mother plant) and (b) the effect of osmotic potential, and salinity stresses on seed germination of Egyptian crowfoot grass.

## 2. Results

### 2.1. Temperature, storage and light

A significant three-way interaction among temperature, storage, and light was detected, along with significant two-way interactions between temperature × storage and temperature × light on seed germination (Table 1).

The highest germination percentage was observed at 30 °C across all storage times (Fig 1). Seeds stored for two months exhibited consistently lower germination than the longer storage times (five and 12 months) at all constant temperatures (Fig 1). Maximum germination (52.5%) occurred after 12 months of storage at 30 °C under light/dark conditions, whereas no germination was detected at 5, 10, and 40 °C. Notably, germination of seeds stored for two months at 25 °C under light/dark conditions (31%) did not differ significantly from that of seeds stored for 12 months at 15 °C (30%).

Egyptian crowfoot grass germination was significantly influenced by temperature, storage and light (Fig 1). When exposed to light, seed germination was consistently higher than in darkness across all constant temperatures at 5 and 12 months storage than 2 months (Fig 1). Under light conditions, germination ranged from 16 to 52.5%, depending on

**Table 1. Analysis of variance for seed germination of Egyptian crowfoot grass (*Dactyloctenium aegyptium*) at constant temperature combined with storage times and light conditions.**

| Source of variation | df | MS | F-Value | P-Value |
|---|---|---|---|---|
| Temperature | 7 | 8227.01 | 911.57 | ** |
| Storage | 2 | 1792.3 | 198.59 | ** |
| Light | 1 | 18741.40 | 2076.58 | ** |
| Temperature × storage | 14 | 223.90 | 24.81 | ** |
| Temperature × light | 7 | 1788.70 | 198.19 | ** |
| Light × storage | 2 | 15.85 | 1.75 | NS |
| Temperature × storage × light | 14 | 18.10 | 2.01 | ** |
| Error | 336 | 9.00 | | |
| C.V. (%) | 21.75 | | | |

**indicate significance at the 0.01 probability level.

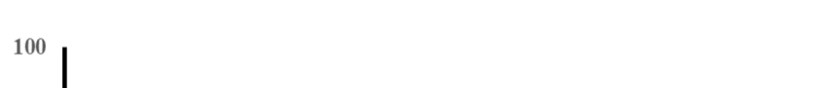

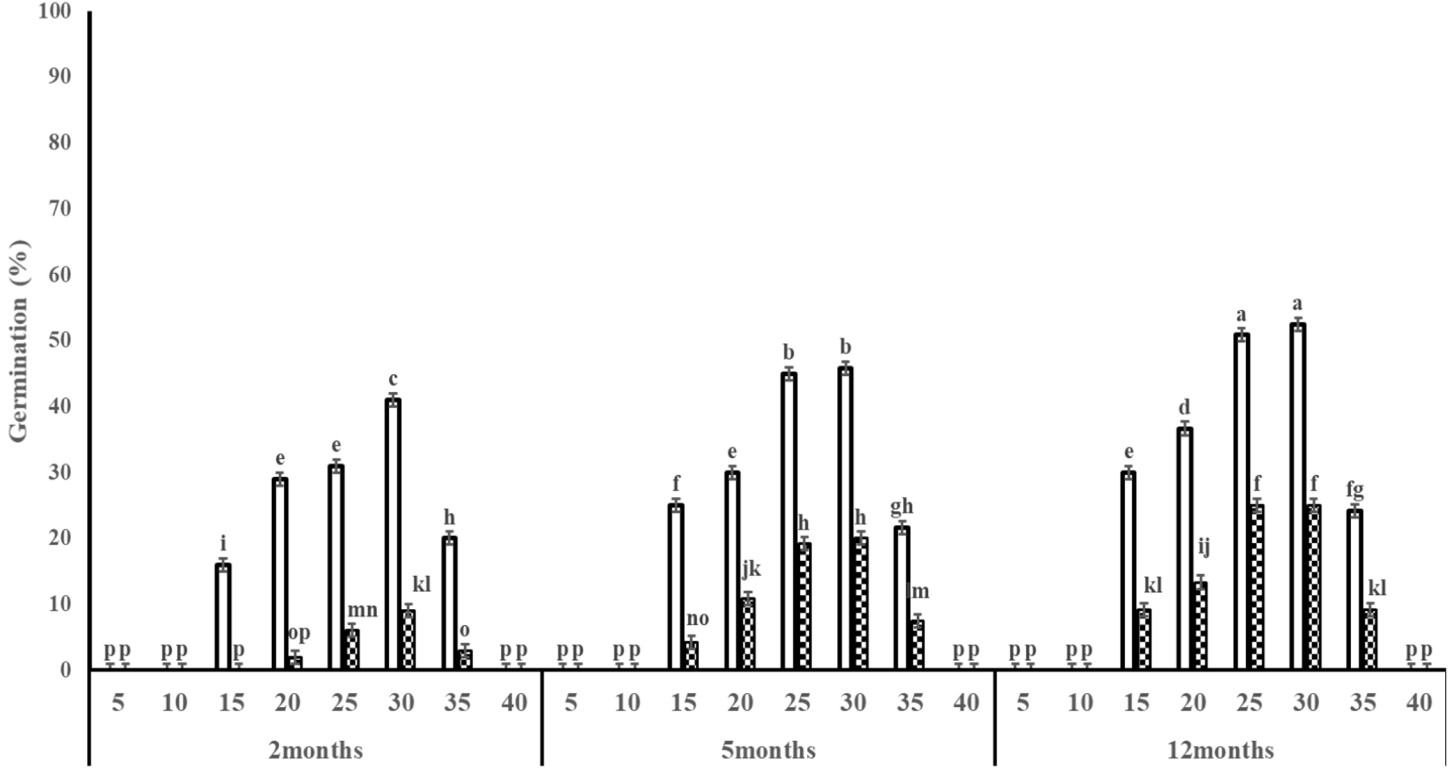

**Fig 1. Effect of interaction constant temperatures, storage times and light regimes on seed germination of Egyptian crowfoot grass.** Nails on the vertical bars represent the standard error of the means.

temperature and storage times. In contrast under dark conditions germination was much lower, ranged from 2 to 25% that was the difference statistically significant (Fig 1). Based on the small seed size, light appears to be the predominant factor regulating germination in this weed species. Also Temperatures ranging from 25 to 30 °C can significantly enhance the germination of this weed. Given the critical role of temperature in regulating germination, it is evident that conditions within this thermal range are particularly conducive to germination success.

with increasing seed storage times in light/dark regime, the maximum germination percentage improved, and the time required to reach maximum germination was shorter compared to seeds stored for two months (Fig 2 and Table 2). For seeds stored two months of storage, estimating the confidence limits of the averages revealed that the maximum percentage of seed germination at different constant temperatures was significant differences from each other (P ≤ 0.05).

In comparison, seed germination exhibited a significant differences across constant temperatures for five months of storage times (P ≤ 0.05). However Seed germination at 25 and 30 ºC after 12 months of storage showed no significant different (P ≤ 0.05). notably the highest germination percentages were recorded for seeds stored for 12 months (Table 2).

As shown in Table 2, there was no significant difference in the $T_{50}$ value between temperatures 15 and 25 ºC, or between 20 and 30 ºC after 2 months of storage times. Although, the $T_{50}$ value at 35 ºC differed significantly from all other constant temperatures. For storage times of five and 12 months, the estimation of confidence limits for the $T_{50}$ values revealed significant difference across all constant temperatures (P ≤ 0.05).

The duration of the lag phase at 30, 25, and 20 ºC was shorter than at 15 and 35 ºC, especially at 12 months of storage compared to 5and 2 months storages (Fig 2- sigmoid equation). At 35 ºC, the onset of germination occurred 12 days after seed imbibition for storage times of two and five months, whereas seed germination was observed within less than five days for seeds stored for 12 months. The time required to reach maximum germination for seeds stored for 12 months at 30 °C was less than 10 days, compared to more than 20 days for seeds stored for two and five months (Fig 2).

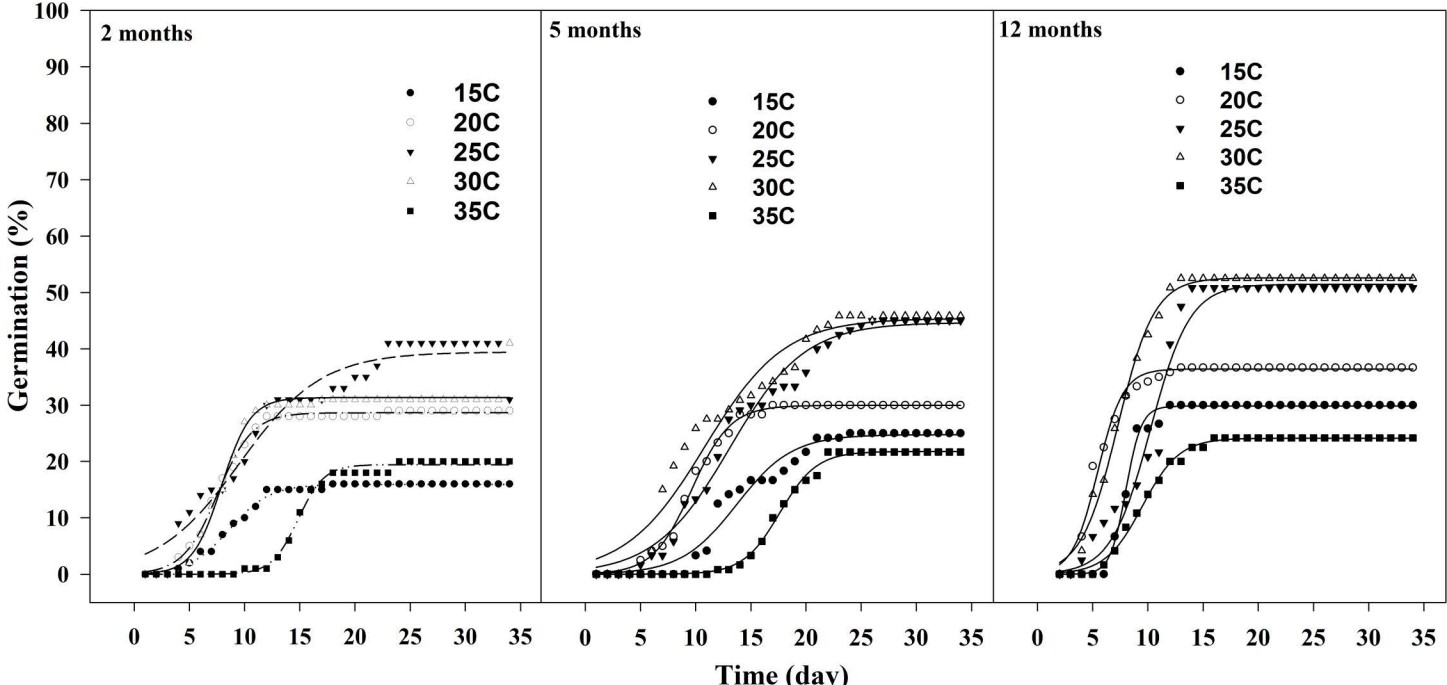

**Fig 2. Effect of constant temperatures and storage times on cumulative germination of Egyptian crowfoot grass in light/dark regime.**

**Table 2. Estimated parameters of the three-parameter sigmoidal equation for cumulative germination in storages times and constant temperatures.**

| Storage (month) | Temperature | $G_{max}$ (%) | $G_{rate}$ | $T_{50}$ (day) | $R^2$ |
|---|---|---|---|---|---|
| 2 | 15 | 15.93 (±0.12*) | 1.82 (±0.08) | 8.63 (±0.18) | 0.99 |
|  | 20 | 28.64 (±0.14) | 1.56 (±0.06) | 7.56 (±0.07) | 0.99 |
|  | 25 | 39.44 (±0.90) | 3.70 (±0.40) | 9.55 (±0.44) | 0.94 |
|  | 30 | 31.35 (±0.40) | 1.13 (±0.15) | 7.91 (±0.17) | 0.97 |
|  | 35 | 19.44 (±0.17) | 1.19 (±0.08) | 14.84 (±0.09) | 0.98 |
| 5 | 15 | 24.75 (±0.48) | 2.51 (±0.25) | 13.72 (±0.30) | 0.97 |
|  | 20 | 29.97 (±0.15) | 1.75 (±0.06) | 9.63 (±0.07) | 0.99 |
|  | 25 | 44.65 (±0.72) | 3.39 (±0.23) | 13.06 (±0.27) | 0.98 |
|  | 30 | 45.38 (±1.06) | 3.58 (±0.38) | 10.85 (±0.43) | 0.95 |
|  | 35 | 21.77 (±0.16) | 1.66 (±0.07) | 17.63 (±0.08) | 0.99 |
| 12 | 15 | 29.84 (±0.17) | 0.70 (±0.05) | 8.00 (±0.05) | 0.99 |
|  | 20 | 36.41 (±0.26) | 1.15 (±0.08) | 5.45 (±0.09) | 0.98 |
|  | 25 | 51.42 (±0.67) | 1.78 (±0.16) | 10.23 (±0.19) | 0.97 |
|  | 30 | 52.58 (±0.27) | 1.65 (±0.07) | 7.27 (±0.08) | 0.99 |
|  | 35 | 24.12 (±0.13) | 1.66 (±0.07) | 9.49 (±0.08) | 0.99 |

Abbreviation: $G_{max}$ is the maximum germination (%), $T_{50}$ is the times (day) required to reach 50% of the maximum germination, and $G_{rate}$ indicates the slope.

*The numbers in parentheses are standard errors.

At 35 ℃, seeds stored for two and five months had the highest $T_{50}$ values (14 and 17 days respectively). In contrast, the $T_{50}$ it was nine days for seeds that were stored for 12 months at 35 ℃. At all storage treatments, a minimum of $T_{50}$ was estimated for incubated seeds at 20 ℃ (5.45–7.56 days) (Table 2). Also, the maximum cumulative germination percentage in the seeds stored for 12 months was higher than in other storage times (Fig 2 and Table 2).

## 2.2. Alternating temperatures, storage and light

No significant interaction effects were observed among the three experimental treatments or between temperature and storage conditions on germination. Interaction effects temperature × light and light × storage were significant on seed germination of Egyptian crowfoot grass (Table 3).

**Table 3. Analysis of variance for seed germination of Egyptian crowfoot grass (*Dactyloctenium aegyptium*) at alternating temperatures combined with storage times and light conditions.**

| Source of variation | df | MS | F-Value | P-Value |
|---|---|---|---|---|
| Temperature | 2 | 7691.3 | 259.58 | ** |
| Storage | 2 | 1216.6 | 41.06 | ** |
| Light | 1 | 68503.7 | 2312.00 | ** |
| Temperature × storage | 4 | 20.1 | 0.68 | NS |
| Temperature × light | 2 | 701.0 | 23.66 | ** |
| Light × storage | 2 | 142.8 | 4.82 | ** |
| Temperature × storage × light | 4 | 36.1 | 1.22 | NS |
| Error | 90 | 29.6 |  |  |
| C.V. (%) | 10.99 |  |  |  |

**indicate significance at the 0.01 probability level.

As shown in Fig 3A, seed germination under continuous darkness was consistently reduced compared to the light/dark regime across all alternating temperatures. The highest seed germination (96%) was recorded at 30/20 °C under light conditions (Fig 3A). In the light/dark regime, seed germination was 68% at 35/25 °C and 59% at 25/15 °C. Notably, for seeds incubated in complete darkness, germination were significantly lower, with 36% at 30/20 °C, 20% at 35/25 °C, and 16% at 25/15 °C.

Germination under light/dark regime (69%, 76%, 77%) significantly exceeded (p<0.01) that under continuous darkness (16%, 24%, 32%) across all storage periods (2, 5, 12 months) (Fig 3B). Twelve months of storage under continuous darkness resulted in a 100% increase in germination compared to the 2-month storage times (Fig 3B).

Across all storage times, the cumulative seed germination at 30/20 exceeded 95%, while at 35/25 °C and 25/15 °C, the values ranged from 67% to 71% and 52% to 65%, respectively (Fig. 4). Cumulative germination analysis revealed that the lag phase for seeds stored for two months was longer than for those stored for five and 12 months at all tested alternating temperatures (Fig 4).

According parameter sigmoidal equation in light/dark regime treatment, shown in Table 3, seed germination was highest (95.30–98.56%) at the alternating temperature of 30/20 °C. The difference in germination percentage at this temperature was statistically significant compared to other alternating temperatures, as indicated by the estimated confidence limits (P ≤ 0.05) (Table 4).

The time to achieve 50% maximum germination ($T_{50}$) ranged from 5 to 20 days for seeds stored 2–12 months under 30/20°C alternating temperatures (Table 4). $T_{50}$ values for 12-month-stored seeds were significantly reduced (15 days shorter; P<0.05) compared to those stored for 2 or 5 months. At 2 months storage, $T_{50}$ reached 20.90, 17.55, and 20.00 days under 35/25°C, 30/20°C, and 25/15°C regimes, respectively. Corresponding values for 5month storage were 11.41, 9.98, and 7.31 days, decreasing further to 5.58, 5.39, and 6.46 days after 12 months storage (Table 4).

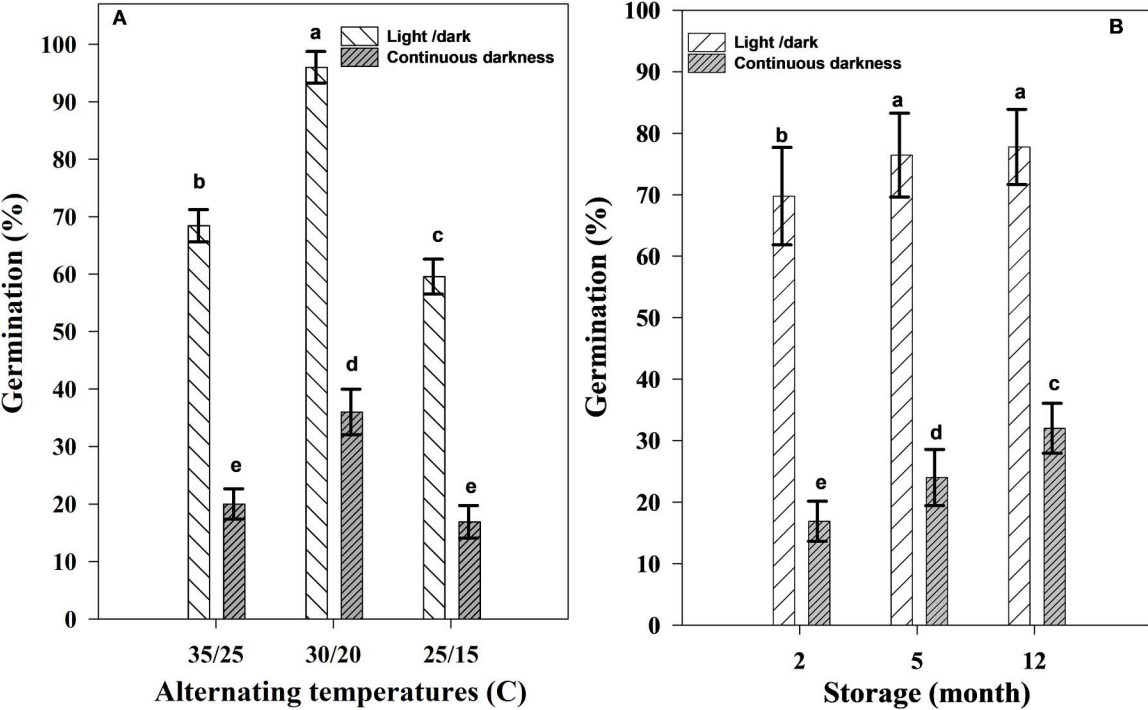

**Fig 3. Effect of alternating temperatures (A) and storage times (B) under light and dark regimes on seed germination of Egyptian crowfoot grass.** Nails on the vertical bars represent the standard error of the means.

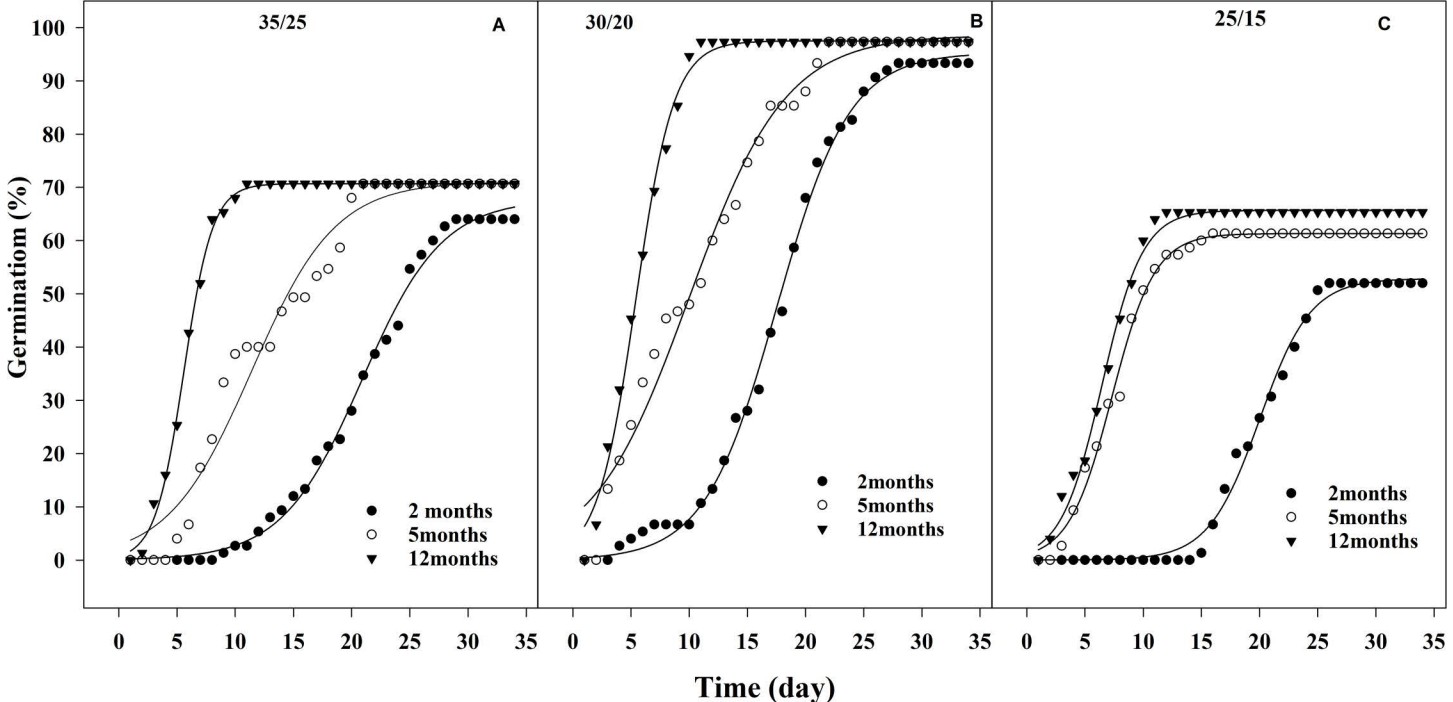

**Fig 4. Effect of alternating temperatures (35/25 ºC(A), 30/20 ºC (B), and 25/15 ºC (C)) and storage times on cumulative germination of Egyptian crowfoot grass in light/dark regime.**

**Table 4. Estimated parameters of three-parameter sigmoidal equation for cumulative germination in storage and alternating temperatures.**

| Temperature (ºC) | Storage (month) | $G_{max}$ | $G_{rate}$ | $T_{50}$ | $R^2$ |
|---|---|---|---|---|---|
| 35/25 | 2 | 67.77 (±1.02*) | 3.44 (±0.14) | 20.90 (±0.18) | 0.99 |
| | 5 | 71.06 (±1.45) | 3.59 (±0.32) | 11.41 (±0.37) | 0.96 |
| | 12 | 70.69 (±0.20) | 1.23 (±0.03) | 5.58 (±0.04) | 0.99 |
| 30/20 | 2 | 95.30 (±0.81) | 3.10 (±0.40) | 17.55 (±0.12) | 0.99 |
| | 5 | 98.56 (±1.33) | 4.15 (±024) | 9.98 (±0.26) | 0.98 |
| | 12 | 97.53 (±0.41) | 1.64 (±0.06) | 5.39 (±0.06) | 0.99 |
| 25/15 | 2 | 52.85 (±0.68) | 2.19 (±0.12) | 20.00 (±0.14) | 0.99 |
| | 5 | 61.32 (±035) | 1.84 (±0.08) | 7.31 (±0.09) | 0.99 |
| | 12 | 65.69 (±0.30) | 1.75 (±0.06) | 6.46 (±0.07) | 0.99 |

Abbreviation: $G_{max}$ is the maximum germination (%), $T_{50}$ is the times (day) required to reach 50% of the maximum germination, and $G_{rate}$ indicates the slope.

*The numbers in parentheses are standard errors.

## 2.3. Salinity and osmotic potential

Based on the fitted three-parameter logistic equation for seed germination at different NaCl concentrations, the highest seed germination (95%) was observed in the control treatment (0 mM NaCl) (Fig 5). Germination declined with increasing salinity, and at 300 mM NaCl, germination was completely inhibited. seed Germination decreased to 68% and 23% at

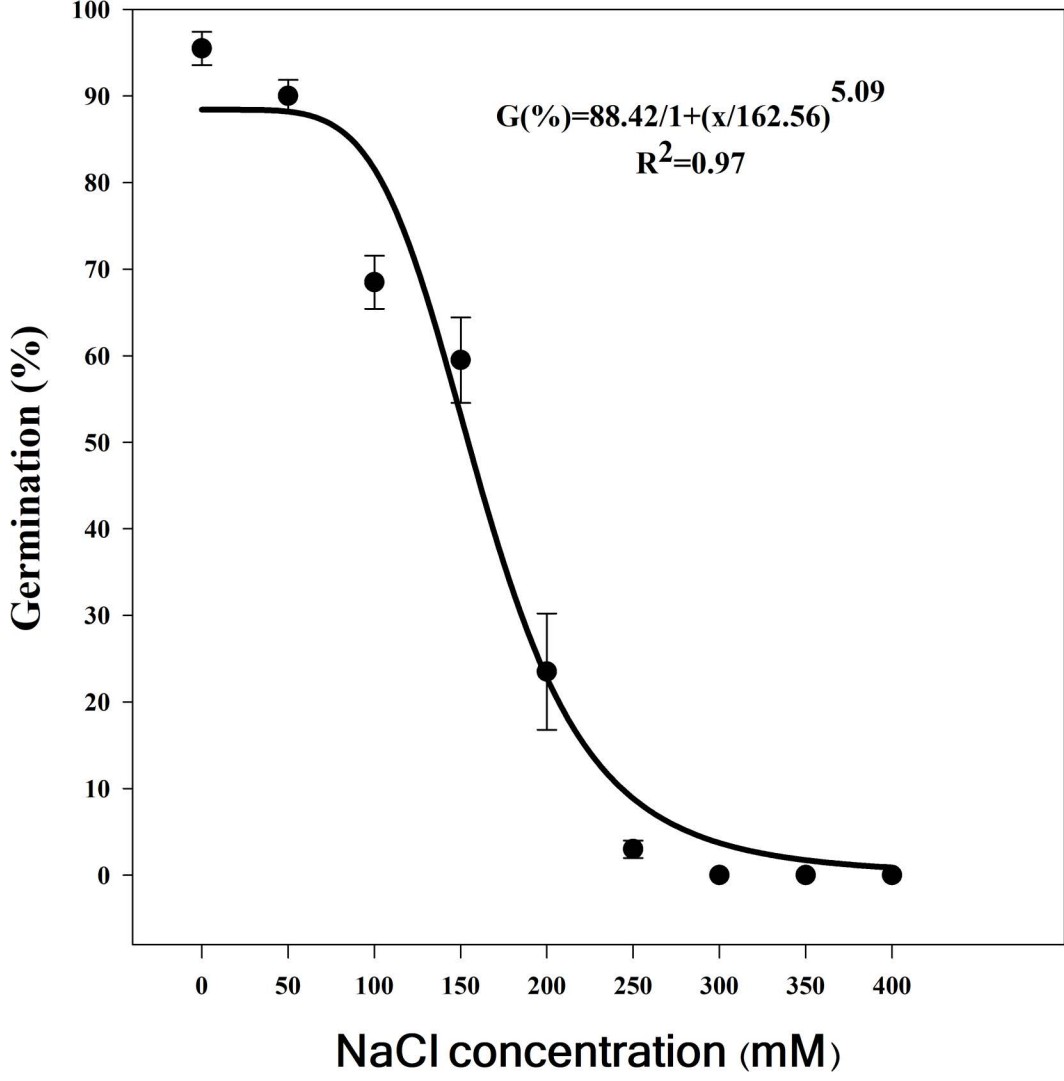

**Fig 5. Effect of NaCl concentration (mM) on seed germination of Egyptian crowfoot grass.** The bold line represents a three-parameter logistic model fitted to the data. Vertical bars represent the standard error of the mean.

NaCl concentrations of 100 mM and 200 mM, respectively, while at 50 mM NaCl, germination remained high at 90%. The NaCl concentration required to induce a 50% reduction in seed germination ($X_{50}$) was estimated at 162 mM. Germination of Egyptian crowfoot grass decreased sharply as NaCl concentrations exceeded 150 mM, with notable seed germination of 23% and 3% at 200 and 250 mM NaCl, respectively.

With increasing drought stress (i.e., a decrease in osmotic potential) from 0 to −0.6 MPa, seed germination decreased from 90% to 45% (Fig 6). At an osmotic potential of −0.8 MPa, only 5% of seeds germinated, while germination was completely inhibited at −1 MPa (Fig 6). The osmotic potential required for 50% inhibition of maximum germination was −0.49 MPa. A 30% reduction in germination was observed when the osmotic potential decreased from −0.2 to −0.4 MPa, while a 40% reduction occurred as the osmotic potential decreased from −0.6 to −0.8 MPa.

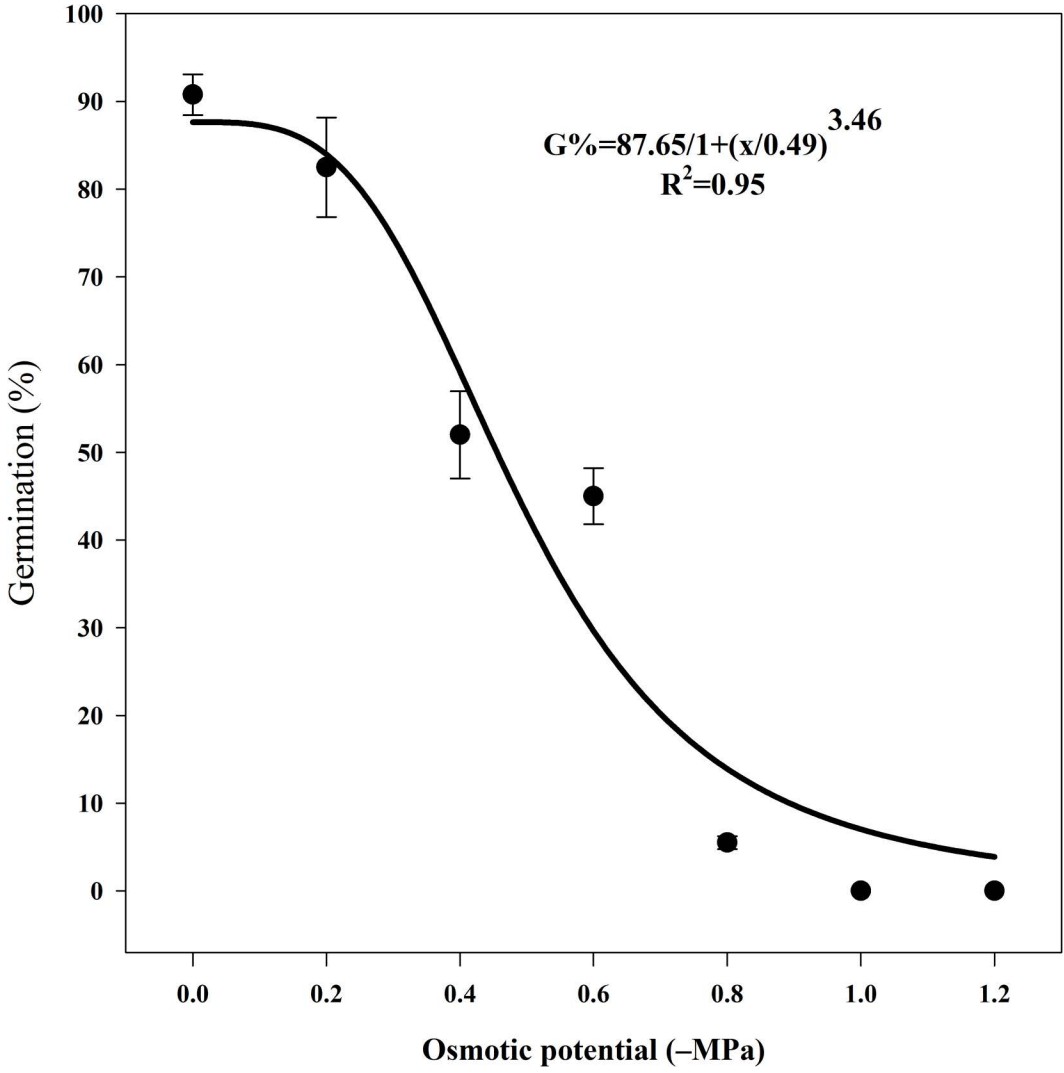

$$G\%=87.65/1+(x/0.49)^{3.46}$$
$$R^2=0.95$$

**Fig 6. Effect of osmotic potential (-MPa) on seed germination of Egyptian crowfoot grass.** The bold line represents a three-parameter logistic model fitted to the data. Vertical bars represent the standard error of the mean.

## 3. Discussion

### 3.1. Constant, alternative temperature, light and storage

The results of constant and alternative temperatures demonstrated, the seeds were exposed to light, germination was significantly higher than in darkness at all temperatures. Under the light/dark regime, there was a statistically significant difference between 30 °C (46%) and other constant temperatures: 15 °C (23%), 20 °C (31%), and 35 °C (21%). In the continuous darkness treatment, seed germination ranged from 4 to 18% depending on the temperatures (Fig 2b). Maximum germination occurred under light conditions at 30/20°C alternative temperatures. These results indicate that under Khuzestan's climate, Egyptian crowfoot grass exhibits peak germination during May (32.3/22.7 C day/night)-June (38.3/31.1 C day/night) when average temperatures approximate this optimal regime. Conversely, July's elevated temperatures (exceeding 35°C daytime) significantly suppress germination rates compared to June conditions. Storage times also

revealed that longer periods led to higher germination percentages, and the time required to reach maximum germination, particularly under alternating and constant temperatures was shorter in the 12 months treatment compared to the 5 and 2 month treatments. This suggests that seeds rain filled from the mother plant in the previous year and remaining in the soil for about one year germinate more rapidly upon receiving the necessary temperature and light cues. Therefore, the timing of application in weed management programs may also be influenced.

The findings indicate that, alongside temperature- a well-documented driver of weed seed germination—light also plays a critical role, particularly in small-seeded weed species. The results of Constant and alternative temperatures revealed, the seed germination of Egyptian crowfoot grass is stimulated by light, indicating that the species is positively photoblastic. Gibberellin production, regulated by phytochrome, plays a crucial role in mediating plant responses to light. Exposure to red light converts phytochrome from its inactive Pr form which inhibits germination into the active Pfr form, promoting germination. In contrast, far-red light reverses this process, reverting Pfr back to the inhibitory Pr state [29].

High temperatures suppress seed germination in many plant species, primarily due to elevated levels of endogenous abscisic acid (ABA). This increase in ABA is mediated by the transcriptional activation of genes involved in ABA signaling pathways [30]. Also Under high-temperature conditions, seed sensitivity to abscisic acid (ABA) increases, partly because the embryo is unable to deactivate ABA and because dissolved oxygen levels decline in water-imbibed seeds [31].

The findings indicate the widespread prevalence of this weed species throughout Khuzestan from early to late spring, demonstrating significant invasive potential in direct-seeded rice (DSR) fields, sugarcane, and urban green spaces. The photoblastic nature of Egyptian crowfoot grass carries ecological significance, as light serves as a soil depth indicator that promotes higher germination rates in surface-positioned seeds compared to deeply buried ones [32].

Small-seeded species such as Egyptian crowfoot grass may possess insufficient seed reserves to support emergence from deeper soil layers. Other potential factors limiting germination of deeply buried seeds include light deprivation, hypoxia (oxygen deficiency), and reduced gaseous diffusion rates at greater depths [2,33]

The findings suggest that agricultural practices involving shallow burial of weed seeds may enhance seedling emergence. In no-tillage systems, where seeds remain predominantly on the soil surface, a greater density of emerged seedlings is observed. This early emergence promotes premature weed-crop competition, necessitating timely intervention in integrated weed management strategies [34]. Conversely, deep tillage reduces light penetration to weed seeds, thereby limiting germination and contributing to a decline in the soil seed bank. Nevertheless, a thorough understanding of the seed bank's history is essential. Previous deep plowing may have sequestered substantial quantities of weed seeds, which could become re-exposed through subsequent soil disturbance. Mitigating this risk requires insights into the persistence and viability of weed seeds within the soil matrix [35].

An additional validated weed management strategy employs plant-derived mulches within integrated cropping systems. In Khuzestan Province's predominant wheat-rice rotation system, conservation tillage with wheat residue retention provides multifunctional suppression through: 1-Physical suppression of seedling emergence by acting as a surface barrier; 2-Reduction in photosynthetic photon flux density (PPFD), which constrains phytochrome-regulated germination in photoblastic weed species; and 3- Gradual release of allelochemicals, which interfere with root development and cellular processes in emerging weed seedlings [36–38].

False or stale seedbed techniques represent ancillary strategies within integrated weed management frameworks [39]. This approach entails precisely timed tillage during periods when diurnal temperatures range 30/20 °C (day/night), facilitating the upward redistribution of weed seeds to the soil surface. Subsequent irrigation and light exposure induce germination particularly those seeds that have persisted in the soil seed bank for prolonged durations, allowing for the preemptive elimination of weed seedlings through mechanical means (e.g., disking, flaming) or chemical control using non-selective herbicides such as paraquat or glyphosate. By targeting early-stage weeds before crop establishment, the false or stale seedbed techniques substantially depletes the soil seed bank and supports long-term weed suppression [40].

The results of the effect of constant temperatures on seed germination of Egyptian crowfoot grass (collected from the United States) indicate the maximum germination occurred at 30 ºC (42%). additionally, seed germination at 35, and 15 ºC, was lower than 20, 25, and 30 ºC temperatures, when seeds were stored for six months at 28 ºC [25].

Additionally, after ripening due to storage has also had a positive effect on seed germination. For instance, the American and Philippine populations exhibited maximum seed germination at alternating day/night temperatures of 25/15 °C (92%) and 30/20 °C (94%), respectively [2,25]. In contrast, the Iranian population showed the highest seed germination (97.33%) at 30/20 °C after 12 months of storage, while the lowest rate (52.85%) was observed at 25/15 °C after two months of storage times. Germination of freshly harvested seeds was stimulated by both light and storage, suggesting that after-ripening is a prerequisite for germination in this species. However, the response to after-ripening varies significantly among species. For example, seeds of *Echinochloa colona* (L.) are stimulated by light but are not influenced by storage times [41]. Seed germination after 5 and 12 months of storage was higher than after 2 months under both light and dark regimes (Fig 3b). Specifically, seeds stored for 12 months exhibited the highest germination (77%) under light conditions, compared to those stored for 5 and 2 months (76% and 69%, respectively) (Fig 3b). Increasing the storage times from 2 to 12 months enhanced seed germination from 16% to 32% in darkness, while germination under the light/dark regime ranged from 69% to 77% (Fig. 3b).

Similarly, seeds of Egyptian crowfoot grass from the United States, stored at room temperature (28 °C) for 6 months, exhibited higher seed germination (94.7%, 86.6%, and 86.3%) when exposed to alternating temperatures of 35/20, 30/20, and 30/15 °C, compared to 25/10 °C (67.8%) [25]. The time required to reach maximum germination was less than 3 days at higher alternating temperatures. In contrast, seed germination in a population collected from the Philippines was higher at 25/15 °C (92%) than at 30/20 °C (70%) and 35/25 °C (44%). It can be concluded that seed germination significantly decreased (17–31%) under dark conditions across the three alternating temperatures [2]. Chauhan and Johnson (2009a) suggested that goosegrass seeds lose sensitivity to light after three months of after-ripening. The germination responses of Egyptian crowfoot grass are similar to those of other summer grass weed species, such as liver seedgrass (*Urochloa panicoides* P. Beauv.) and junglerice (*E. colona*), whose seeds germinate at alternating temperatures ranging from 25/15 °C to 35/25 °C [41–43].

Research on eight *Euphorbia* species demonstrated that seed germination percentage increased with longer storage times. Seeds stored for more than 150 days showed significantly higher germination rates compared to those stored for fewer than 150 days. Additionally, the study found that freshly harvested seeds of *Eriochloa villosa* (Thunb.) exhibited lower germination potential, germination rate and germination index than stored seeds [44].

The effects of dry storage on *Isatis violascens* seed germination revealed distinct temporal patterns. At storage initiation (time zero), seed germination were minimal, with only 1% of silicles and 3% of isolated seeds viable. After six months of dry storage under optimal conditions (darkness, 2–5°C), germination rates increased to 11% for silicles and 24% for isolated seeds. Continued after-ripening between 6–12 months significantly enhanced germination capacity, reaching 38% for silicle-contained seeds and 71% for isolated seeds by the end of the 12-month storage times [45].

The research of seven temperatures (10–40°C at 5°C intervals) and seven after-harvest periods (30–540 days after harvest) revealed distinct temperature-dependent germination responses among nine *Amaranthus* species, with most exhibiting optimal germination (70–100%) at 35°C, while negligible germination occurred at 10–15°C for *A. albus, Amaranthus. deflexus, Amaranthus. graecizans*, and *Amaranthus. lividus*. Notably, *Amaranthus. cruentus, Amaranthus. hybridus*, and *Amaranthus. retroflexus* showed intermediate tolerance, germinating at 15°C (>60%) In contrast, *Amaranthus. blitoides* and *Amaranthus. viridis* uniquely required alternating temperatures (30/10 °C) and light exposure to achieve high germination (>90%), underscoring their adaptation to fluctuating environmental conditions. After-ripening duration significantly influenced germination in *A. cruentus, A. hybridus*, and A. *retroflexus*, but had marginal effects on *A. graecizans and A. lividus*, highlighting species-specific dormancy strategies [46].

## 3.2. Salinity and osmotic potential

Germination percentages declined with increasing salinity: 90% at 50 mM, 68% at 100 mM, 59% at 150 mM, and 23% at 200 mM. Based on the logistic equation, the $X_{50}$ (salinity reducing maximum germination by 50%) was estimated at 162 mM. Germination of Egyptian crowfoot grass decreased sharply at NaCl concentrations exceeding 150 mM, indicating that this species is more tolerance to salinity than other grasses, such as *E. colona* (106 mM), *Eleusine indica* (L.) Gaertn. (74 mM) and *Leptochloa chinensis* (L.) Nees (50 mM) [19].

High salinity reduces the osmotic potential of water in the soil, ultimately leading to decreased water absorption by dry seeds (imbibition process). Additionally, excessive uptake of sodium ($Na^+$) and chloride ($Cl^-$) ions from the soil causes ionic stress and toxicity, thereby disrupting biochemical processes such as nucleic acid and protein metabolism, energy production, and respiration [47]. Salinity disrupts the balance of key nutrients and hormones particularly gibberellin (GA) and abscisic acid (ABA) during seed germination. High salinity levels can delay germination or, depending on the plant's salt tolerance, completely inhibit it. Under severe salt stress, the delicate balance between reactive oxygen species (ROS) production and scavenging is disturbed, leading to the accumulation of harmful ROS such as hydroxyl radicals, superoxide, and hydrogen peroxide. These ROS oxidatively damage critical macromolecules (e.g., proteins, carbohydrates, nucleic acids, and lipids) and disrupt cellular structures (e.g., membranes), ultimately impairing or preventing germination [48].

Limited information is available on the response of this weed to salt stress. Egyptian crowfoot grass seeds exhibited varying germination behaviors across different populations. For instance, when exposed to seawater concentrations, the Otta population showed a gradual decline in germination, with the lowest seed germination observed at 100% seawater concentration. In contrast, the Lagos population exhibited a sharp decline in germination, with complete inhibition at 20% seawater concentration [26]. Similarly, under salinity conditions, less than 20% of *Brachiaria eruciformis* seeds germinated at 150 mM NaCl [49]. Japanese brome (*Bromus japonicus*) demonstrated higher salinity tolerance, with an $X_{50}$ value predicted at 202 mM NaCl [50]. According to Holm et al. (1977), the ability of this plant to produce seeds suggests that its germination capacity could significantly contribute to the establishment of dynamic populations in saline environments [3].

The osmotic potential required for 50% inhibition of maximum germination was −0.49 MPa, although some seeds germinated at −0.8 MPa. However, no germination occurred when the osmotic potential was greater than −0.8 MPa. In comparison, the osmotic potential for 50% inhibition of maximum germination in the Philippine population was −0.23 MPa [2]. Notably, no germination occurred at −0.8 ≤ MPa for American and Philippine populations [2,25]. An osmotic potential greater than −0.8 MPa completely inhibits the germination of both grass and broadleaf weeds [24], while osmotic potential ranging from −0.09 to −0.32 MPa reduces weed germination by 50% compared with the unstressed condition [51] The ability of the Iranian population to germinate at −0.8 MPa suggests greater drought tolerance compared to the American and Philippine populations.

Soil moisture influences both the timing of weed emergence and the number of weed seedlings emerging. In dry conditions, insufficient water can be the primary constraint on seed germination [52]. Water potential is a primary driver of water uptake, including seed imbibition; drought stress reduces water potential, thereby limiting uptake [53].

These findings suggest that seeds of certain species remain dormant until sufficient moisture becomes available, a mechanism that accounts for the mass emergence of seedlings at the onset of the rainy or monsoon season, which typically aligns with the start of the cropping period. This behavior likely reflects an adaptive strategy to delay germination until favorable conditions arise, thereby enhancing seed longevity within the seed bank [19].

Finally, this ability and tolerance to salinity and osmotic potential allows this weed to have a distinct competitive edge over crops like rice (direct seeded), sugarcane and urban green space, which may be subjected to stress in such environments.

## 4. Conclusion

In summary, the present study demonstrated that this weed exhibits distinct germination responses to constant temperatures, alternating temperatures, light, and storage times. Maximum germination (97%) was achieved under light conditions at 30/20°C alternating temperatures after 12 months of storage, confirming its photoblastic nature. Increased storage times (after-ripening) following rainfall from the mother plant significantly reduced the time required to reach peak germination, indicating potential for rapid early-season establishment. Based on the $X_{50}$ parameter, which quantifies the salinity and drought thresholds required to suppress germination by 50%, this weed exhibits considerable tolerance to both stress conditions. Germination persists even at salinity 150 mM (60%) and an osmotic potential of −6 bars (45%). Given climate change-induced shifts and effect on salinity, and drought stress in Khuzestan Province, expansion of this weed in summer crops is highly probable. Consequently, integrated management strategies including deep tillage, wheat straw mulching, and false or stale seedbed techniques should be implemented to deplete its soil seed bank.

## 5. Materials and methods

### 5.1. Seed collection and storage

Seeds of Egyptian crowfoot grass were obtained from Agricultural Sciences and Natural Resources University of Khuzestan located in the Mollasani district of Bavi County, Iran (31°35′56.1″N 48°53′18.8″E) during autumn 2017. Mature seeds were collected from a minimum of 100 individual plants to ensure genetic diversity and representativeness. After collection spikes were air-dried at 25 °C under ambient conditions for one week, then threshed and sieved to separate the grains seeds were cleaned, and placed in a paper bag for storage, allowing them to dry until reaching 11% moisture content and finally stored at room temperature (25 ºC) for two, five, and twelve months before their use in three experiments. No dormancy-breaking treatment was applied; seeds were stored at room temperature for durations specified by the experimental design.

### 5.2. Effect of constant temperatures, storage and light on germination

Eight constant temperatures (5, 10, 15, 20, 25, 30, 35 and 40 ºC) under two light regimes (light/dark (16-h photoperiod/8-h dark) and continuous darkness) and 3 storage times (2, 5 and 12 months after rainfall of mother plants) were tested. A light intensity of 185 µmol m$^{-2}$ s$^{-1}$ was maintained in the germination chambers using fluorescent lamps.

Petri dishes with a diameter of 10 cm, lined with two layers of Whatman filter paper, were used. Thirty seeds were placed in each Petri dish. At the beginning of the experiment, 5 mL of distilled water was added to Petri dishes, and seed germination was counted daily until 34 days with visible protrusion of the radicle to (2 ≥ mm) used as the criterion germination [54]. To create darkness, Petri dishes were wrapping in two layers of aluminum foil. To maintain optimal moisture under dark conditions, seeds were fully shielded from white light. Petri dishes were inspected solely under green safe light to preserve complete darkness during observations.

### 5.3. Effect of alternating temperatures, storage and light on seed germination

Three alternating temperatures 35/25, 30/20, and 25/15 ºC under two light regimes light/dark and continuous darkness) and 3 storage times (2, 5 and 12 months after rainfall of mother plants) were selected (The selected temperature regimes reflect the thermal variations observed in Khuzestan). The experimental procedure was similar to that used for constant temperatures.

### 5.4. Effect of osmotic potential and salinity on seed germination

Seven concentrations of polyethylene glycol 6000 (0, −0.2, −0.4, −0.6, −0.8, −1, and 1.2 MPa) were assessed to determine the effect of osmotic potential on seed germination. The solution was prepared by dissolving polyethylene glycol 6000 in distilled water based on 25 C temperature [55].

Also, nine concentrations of sodium chloride (NaCl) (0, 50, 100, 150, 200, 250, 300, 350, and 400 mM) were selected to determine the response of seed germination to salt stress. Thirty seeds were placed in each Petri dish, and 5 mL of the respective solutions were added to each dish. Petri dishes covered with a plastic bag to prevent water loss. Petri dishes were placed at 30/20 ℃ (optimal temperature based on alternating temperatures) under the light/dark regime. germination was counted daily for 21 days.

### 5.5. Statistical analyses

A completely randomized design (CRD) was used for all experiments, with four replications per treatment. Prior to conducting ANOVA using SAS software (version 9.4), we verified the assumptions of normality (Shapiro-Wilk test) and homogeneity of variance for each experiment. No data transformation was required as all assumptions were satisfied. A three-way ANOVA was conducted for the two experiments involving constant and alternating temperatures. The factors included temperature, light, and storage times. After ANOVA, means were separated using the least significant difference (LSD) method at probability level of $P \leq 0.05$..

In the light/dark treatment, daily seed counts were recorded, and the relationship between imbibition time and cumulative germination percentage was assessed using regression analysis. For the continuous darkness treatment, where daily counting was omitted to avoid potential errors from accidental light exposure, only the final germination count was taken, followed by an analysis of variance (ANOVA).

The relationship between imbibition time and cumulative germination at different temperatures and storage times under light/dark regime was fitted using a three-parameter sigmoid model in Sigma Plot 14.0. The model (Eq. 1) is defined as follows.

$$G = G_{max}/[(1 + e - (T - T_{50})]/G_{rate}]) \tag{1}$$

Where G is the cumulative of germination (%) at times T, $G_{max}$ is the maximum germination (%), $T_{50}$ is the times (day) required to reach 50% of the maximum germination, and $G_{rate}$ indicates the slope.

Germination percentages from both NaCl and osmotic potential experiments were fitted to a three-parameter logistic model using SigmaPlot software (version 14).

$$G = G_{max}/[1 + (x/x_{50})^\wedge G_{rate}] \tag{2}$$

where, G is the total germination (%) at NaCl concentration or osmotic potential x, $G_{max}$ is the maximum germination (%), $X_{50}$ is the NaCl concentration or osmotic potential for 50% inhibition of the maximum germination (%), and $G_{rate}$ is the slope.

### Author contributions

**Conceptualization:** Ahmad Zare, Elham Elahifard.

**Data curation:** Ahmad Zare, Zahra Asadinejad.

**Formal analysis:** Ahmad Zare.

**Investigation:** Ahmad Zare.

**Software:** Ahmad Zare.

**Visualization:** Ahmad Zare.

**Writing – original draft:** Ahmad Zare, Elham Elahifard.

**Writing – review & editing:** Ahmad Zare, Elham Elahifard.

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
