## [Decision Letter · Decision Letter 0]

16 Jan 2026

PONE-D-25-58145Adaptive Strategies of Egyptian Crowfoot Grass ( Dactyloctenium aegyptium (L.) Willd): Germination Responses to Environmental FactorsPLOS One

Dear Dr. zare,

Thank you for submitting your manuscript to PLOS ONE. After careful consideration, we feel that it has merit but does not fully meet PLOS ONE’s publication criteria as it currently stands. Therefore, we invite you to submit a revised version of the manuscript that addresses the points raised during the review process.

This is a very nice experiment and results. Your interpretations of the data is acceptable. However, your word choosing and writing style needs to be some changes. At lease you can follow some earlier literature written by native speakers. I mostly agree with both reviewers.

Please submit your revised manuscript by Mar 02 2026 11:59PM. If you will need more time than this to complete your revisions, please reply to this message or contact the journal office at plosone@plos.org. Please include the following items when submitting your revised manuscript:

We look forward to receiving your revised manuscript.

Kind regards,

Ahmet Uludag, Ph.D.

Academic Editor

PLOS One

Journal Requirements:

2. We note that your Data Availability Statement is currently as follows: ll relevant data are within the manuscript and its Supporting Information files

4. Please amend the manuscript submission data (via Edit Submission) to include author Zahra Asadinejad

5. Please amend your authorship list in your manuscript file to include author zahra asadinajat

6. No Funding:

Thank you for stating the following financial disclosure:

7. Thank you for stating the following in the Acknowledgments Section of your manuscript:

The author(s) declare that financial support was received for the research, authorship, and/or publication of this article. The current paper is adapted from a research assigned in Agricultural Sciences and Natural Resources University of Khuzestan, with a Grant Number of 981/09 and financially supported by the university; thereby we declare our appreciation for their help.

Additional Editor Comments:

I have added my comments in the file added by a reviewer. I think you can choose different words and style for writing to make your manuscript readable by audience easily

Reviewers' comments:

Reviewer's Responses to Questions

**Comments to the Author**

1. Is the manuscript technically sound, and do the data support the conclusions?

Reviewer #1: Yes

Reviewer #2: Yes

2. Has the statistical analysis been performed appropriately and rigorously? 

Reviewer #1: Yes

Reviewer #2: Yes

3. Have the authors made all data underlying the findings in their manuscript fully available?

Reviewer #1: Yes

Reviewer #2: Yes

4. Is the manuscript presented in an intelligible fashion and written in standard English?

Reviewer #1: Yes

Reviewer #2: Yes

5. Review Comments to the Author

Reviewer #1: Hello, first of all, I wish you a good day and thank you for offering me a review. The article is well-written and understandable. I answered the questions posed here to the best of my knowledge. Other than that, I have nothing to say; this experience was enjoyable. I wish you success, best regards.

Reviewer #2: The article is of great scientific and agronomic interest. All its parts are well-developed; however, the paper is overly descriptive, and parts of the Introduction and Discussion could be shortened. There are too many references, and at times I feel like I'm reading a lengthy review rather than a paper. The extensive number of current (2025) and historical (1990) references demonstrates the authors' broad knowledge of the subject.

6. PLOS authors have the option to publish the peer review history of their article (what does this mean?). If published, this will include your full peer review and any attached files.

Reviewer #1: No

Reviewer #2: **Yes:**yes

---

## [Decision Letter · Decision Letter 1]

9 Mar 2026

Adaptive Strategies of Egyptian Crowfoot Grass ( Dactyloctenium aegyptium (L.) Willd): Germination Responses to Environmental Factors

PONE-D-25-58145R1

Dear Dr. Zare,

We’re pleased to inform you that your manuscript has been judged scientifically suitable for publication and will be formally accepted for publication once it meets all outstanding technical requirements.

Kind regards,

Ahmet Uludag, Ph.D.

Academic Editor

PLOS One

Additional Editor Comments (optional):

I would like to congratulate you that your manuscript is acceptable. Some typhos should br corrected as reviewer suggested. I also recommend you check your text for any other small corrections. On the other hand, could you please check plosone policy on data availability statement and make relevant changes.

Reviewers' comments:

Reviewer's Responses to Questions

**Comments to the Author**

1. If the authors have adequately addressed your comments raised in a previous round of review and you feel that this manuscript is now acceptable for publication, you may indicate that here to bypass the “Comments to the Author” section, enter your conflict of interest statement in the “Confidential to Editor” section, and submit your "Accept" recommendation.

Reviewer #1: All comments have been addressed

Reviewer #3: All comments have been addressed

2. Is the manuscript technically sound, and do the data support the conclusions?

Reviewer #1: Yes

Reviewer #3: Yes

3. Has the statistical analysis been performed appropriately and rigorously? 

Reviewer #1: Yes

Reviewer #3: Yes

4. Have the authors made all data underlying the findings in their manuscript fully available?

Reviewer #1: Yes

Reviewer #3: No

5. Is the manuscript presented in an intelligible fashion and written in standard English?

Reviewer #1: Yes

Reviewer #3: Yes

6. Review Comments to the Author

Reviewer #1: (No Response)

Reviewer #3: The revised manuscript is substantially improved and, in my view, now meets PLOS ONE’s criteria for technical soundness and clarity. The Authors have clarified the experimental design, expanded the description of seed collection and storage, and given a much more detailed account of how constant and alternating temperatures, light regimes, and storage intervals were combined in the factorial germination experiments. The inclusion of three‑way ANOVA tables for constant and alternating regimes, together with fitted three‑parameter sigmoidal models for cumulative germination and three‑parameter logistic models for salinity and osmotic potential responses, makes the statistical approach transparent and appropriate for the questions asked.

The data convincingly support this report's main conclusions. The temperature–light–storage experiments show that Egyptian crowfoot grass is positively photoblastic, that germination is favoured by alternating temperatures (especially 30/20 °C), and that after‑ripening over 12 months leads to higher maximum germination and shorter T₅₀ compared with 2‑ and 5‑month storage. The salinity and osmotic potential assays demonstrate that the Iranian population can still germinate under relatively high NaCl concentrations (with an estimated X₅₀ of about 162 mM) and under moderate water deficit (50% inhibition at about −0.49 MPa, with some seeds germinating even at −0.8 MPa), which is higher tolerance than has been reported for some other grass weeds and for other Dactyloctenium populations. These findings are then sensibly linked to likely emergence windows under Khuzestan climate conditions and to practical implications for integrated weed management in direct‑seeded rice, sugarcane, and urban systems (deep tillage, straw mulching, false/stale seedbeds).

Importantly, the Authors have strengthened the comparative and ecological context. The revised Introduction and Discussion now situate the Iranian population relative to previously studied US, Philippine, and Nigerian populations, explaining how differences in after‑ripening duration, maternal environment, and testing regimes can lead to distinct germination “signatures”. They also relate the observed light dependence and shallow‑burial sensitivity to realistic tillage and residue scenarios, and discuss how increased salinity and drought under climate change may further favour this weed’s expansion and competitiveness. These additions help justify the “adaptive strategies” framing and will be valuable for readers interested in population‑level variation in weed seed ecology.

The manuscript is generally well written and intelligible. Many of the earlier awkward phrasings and repetitions have been removed or corrected in the tracked‑changes version. A few very minor typographical issues remain (for example, stray spaces such as “A50%” instead of “A 50%” in the Abstract, or occasional capitalisation inconsistencies like “Direct Seeded Rice”), but these are easy to fix at proof stage and do not impede understanding. The figures and tables are clear and correspond well to the text. The parameter tables for the sigmoidal and logistic fits (Gmax, T₅₀, X₅₀, slope, R²) are particularly helpful for readers who may wish to use your estimates in modelling or management planning.

The only substantive issue that remains is data availability. The current statement indicates that “datasets generated and/or analyzed … are available from the corresponding author upon request” due to “data protection and participant confidentiality”. This does not align with PLOS ONE’s data policy for purely experimental plant germination studies, where there are no human subjects or sensitive personal data. The raw germination counts (or at minimum, per‑replicate percentages for each treatment combination and time point used for curve fitting), together with the underlying data for the salinity and osmotic potential assays, should be made available either as Supporting Information files or in a public repository, with an updated Data Availability Statement reflecting this. I do not see any ethical or legal barrier to sharing these data.

Beyond this data‑policy point, I have no further major concerns. If you can revise the Data Availability Statement to provide direct access to the underlying datasets, I would support publication of the manuscript.

7. PLOS authors have the option to publish the peer review history of their article (what does this mean?). If published, this will include your full peer review and any attached files.

Reviewer #1: No

Reviewer #3: No

---

## [Editor Report · Acceptance letter]

PONE-D-25-58145R1

PLOS One

Dear Dr. zare,

I'm pleased to inform you that your manuscript has been deemed suitable for publication in PLOS One. Congratulations! Your manuscript is now being handed over to our production team.

Kind regards,

on behalf of

Dr. Ahmet Uludag

Academic Editor

PLOS One